

# Performance and evaluation of remote sensing satellites for monitoring dust weather in East Asia

Yuanyuan Zhang[1], Ning Wang[2], Shuanggen Jin[1,3]

[1]School of Surveying and Land Information Engineering, Henan Polytechnic University, Jiaozuo, 454003, China
[2]Nanjing Xinda Institute of Meteorological Science & Technology, Nanjing, 210000, China
[3]Shanghai Astronomical Observatory, Chinese Academy of Sciences, Shanghai, 200030, China

*Correspondence to*: Shuanggen Jin (sgjin@shao.com)

**Abstract.** Satellite remote sensing provides a unique tool for monitoring dust weather in East Asia in real time and
accurately. However, it is still challenging whether these data can effectively and accurately capture the dynamic process of
dust weather. Meanwhile, capability and performances of different satellite remote sensing products are not clear in
monitoring dust weather. In response to the current problems, this study used PM10 concentration data from environmental
monitoring stations to evaluate the continuity, accuracy and stability of five kinds of satellite remote sensing aerosol
products (FY4A dust score products (DST) and infrared difference dust index (IDDI), MODIS Aerosol Optical Depth
(AOD), Sentinel-5P absorbing aerosol index (AAI) and Himawari-8 AOD) for monitoring and studying dust weather in East
Asia. The results showed that the daily spatial distribution of atmospheric dust presented by the five aerosol products had
good consistency. In particular, the AAI product was not only better than other aerosol products in depicting the continuity
of the spatial distribution of atmospheric dust, but also made up for the inability of other products in obtaining dust
information under the clouds. The ground station PM10 data verification showed that the atmospheric dust POCD of MODIS
AOD, Himawari-8 AOD, Sentinel-5P AAI, FY4A IDDI and DST products during the entire dust weather process were 91%,
35.5%, 24.4%, 15.8% and 14.6 respectively. Under the same observation time and space conditions, the atmospheric dust
POCD of MODIS AOD, Himawari-8 AOD, FY4A IDDI and DST, and Sentinel-5P AAI products were 85.7%, 43.8%,
37.3%, 37.3% and 5.6%, respectively. Overall, the MODIS AOD product performed best in monitoring dust weather in East
Asia with high accuracy, and then the Himawari-8 AOD product.

## 1 Introduction

Dust weather refers to a general term for a weather phenomenon in which wind blows dust and sand from the ground into the
air, making the air turbid (Yang et al.,2008; Wang et al.,2013; Zhao et al.,2018). When dust weather occurs, the dust
particles dispersed into the air by the wind not only have an impact on the ecological environment and human life and health
but also have an important impact on global climate change (Mahowald, 2011). On the one hand, the long-distance transport
and dry and wet deposition of dust particles in the atmosphere provide critical mineral supplements for terrestrial and marine





organisms, playing an irreplaceable role in global biochemical processes and carbon cycle systems (Richon et al., 2018; Mahowald et al., 2005; Shao et al., 2011). On the other hand, the diffusion of dust from the surface into the air not only reduces visibility and air quality but also carries some harmful microorganisms and heavy metals that are extremely harmful to human life and health (Mu et al., 2023; Middleton et al., 2017; Rao et al., 2020). In addition, the dust particles staying in

the atmosphere have significant radiative forcing and climate effects, which can affect the radiation balance of the earth-atmosphere system through direct effects, indirect effects and semi-direct effects and are a key factor causing deviations in global climate change predictions and sensitivity assessments (Huang et al., 2006; Huang et al., 2014; Kok et al., 2023). Therefore, obtaining effective information on dust weather in real time and accurately is a key issue for effective early warning and forecasting of dust weather and studying its environmental and climate effects.

In the past, the dust weather processes were mainly monitored by conventional ground weather stations (Akhlaq et al., 2012; Shao and Dong, 2006). However, large-scale dust weather usually originates in remote desert areas with harsh natural environments. The ground weather monitoring stations are extremely susceptible to the influence of complex natural environments, which makes it difficult to achieve long-term monitoring of dust weather. In addition, limited by various factors, it is difficult to build large-scale, high-density ground stations to monitor the formation, development and

disappearance of dust weather in real-time and long-term (Bao et al., 2023). After the 1970s, with the rapid development of various earth observation technologies, satellite remote sensing has become the main tool for dust weather monitoring and research (Li et al., 2021; Jiao et al., 2021; Akhlaq et al., 2012; Shao and Dong, 2006). First of all, satellite remote sensing technology has the advantages of wide observation range, strong timeliness and high economic benefits, which make up for the shortcomings of ground-based site monitoring methods. Secondly, different satellite sensors have different temporal

resolutions, spatial resolutions, spectral resolutions and radiation resolutions. The dust weather information obtained by integrating multiple satellite observation data is more comprehensive. Finally, the complementary advantages of different imaging modalities can achieve all-weather dust weather information acquisition.

The methods of monitoring atmospheric dust aerosol by satellite remote sensing have been continuously improved in the development, forming two different monitoring methods of dust weather: passive remote sensing and active remote sensing.

Passive remote sensing uses the earth-atmosphere system itself to emit or reflect electromagnetic wave information from natural radiation sources to realize quantitative retrieval of atmospheric dust optical parameters, which is mainly divided into the ultraviolet absorption method, the visible near-infrared method, the thermal infrared method, and the microwave polarization index method (Chen et al., 2014; De et al.,2005; Kaufman et al., 2001; Zhang et al., 2006; Huang et al., 2007). Currently, passive remote sensing products that are widely used in dust weather monitoring and research originate from the

aerosol products of the Terra\Aqua Moderate Resolution Imaging Spectroradiometer (MODIS) and Himawari-8 Advanced Himawari Imager (AHI), the absorbing aerosol index (AAI) products of the Aura Ozone Monitoring Instrument (OMI) and the Sentinel-5P Tropospheric Monitoring Instrument (TROPOMI), the infrared difference dust index (IDDI) products of FY-2 series of satellites Visible Infrared Spin Scan Radiometer(VISSR), the dust score (DST), dust strong index (DSI) and aerosol index (AI) products of the FY-3 series of satellites Visible Infrared Radiometer (VIRR) and the dust detection dataset



(DSD) product of FY- 4A Advanced Geostationary Radiation Imager (AGRI). For example, Filonchyk et al. (2020) integrated MODIS AOD and OMI AAI products to study two severe dust weather processes that occurred in the South Gobi Desert of China. The results showed that the AOD value in the area affected by dust weather exceeded 1, and the AAI value was in the range of 0.7-3.9. Sun et al. (2022) integrated Himawari-8 AOD and FY-4A DSD products to analyze the spatiotemporal distribution and transmission characteristics of two dust events in North China in March 2021. Gao and

Washington (2010) used TOMS AI products to characterize the frequency and intensity of dust weather in the Tarim Basin, and explored the teleconnection between dust activities and the Arctic Oscillation. Ye and Zheng (2023) used the TROPOMI AAI product to study and analyze the impact range of daily dust events from a strong dust weather process that occurred in northern China from March 14 to 18, 2021. Li et al. (2016) analyzed the outbreak, development, transmission and impact range of the Taklimakan desert sandstorm in April 2014 based on the IDDI product of FY-2E and the AI product of FY-3B,

and used the MODIS AOD product to explore the atmospheric dust aerosol load in the areas affected by the sandstorm process. Jiang et al. (2021) used the DSI product of FY-3A to analyze the intensity and seasonal changes of dust weather activities in the Tibetan Plateau from 2010 to 2013.

Active remote sensing relies on artificial radiation sources on the remote sensing platform to emit electromagnetic waves to targets and detect atmospheric dust information by receiving backscattered signals (Liu et al., 2008). So far, CALIOP carried

on the CALIPSO satellite is the most stable, longest-running, most mature and most widely used spaceborne lidar in orbit (Wang et al., 2023). The satellite stopped operating in August 2023 because its fuel reserves had been exhausted and, in its decaying, orbit the satellite can no longer generate sufficient power to operate the science instruments. During its 17 years in orbit, CALIPSO has provided unprecedented measurement data of the vertical structure of the Earth's atmosphere, which has been verified by a large number of ground-based and passive satellite remote sensing observation data (Kim et al., 2018;

Winker et al., 2007; Liu et al., 2018). CALIOP has the advantages of active remote sensing detection and polarization monitoring, which can distinguish dust from complex atmospheric environments. (Liu et al., 2008). Its aerosol classification and vertical structure are currently the most comprehensive and accurate aerosol products and are widely used in the study of atmospheric dust transport, aerosol-cloud interaction and climate effects, etc. (Gui et al., 2022; Jia et al., 2018; Wang et al., 2023). For example, Bao et al. (2023) used CALIPSO data to study the vertical structure of the atmospheric dust layer during

three severe dust weather processes in East Asia, and found that the vertical structure of the dust layer depends on the sources and intensity of the dust weather. Sun et al. (2022) used CALIPSO aerosol products to study and analyze the three-dimensional structure and transmission path of atmospheric dust during the dust weather in North China from March 26 to 30, 2021.

From March 13 to 20, 2021, the strongest dust weather occurred in East Asia in nearly 10 years due to the influence of the

Mongolian cyclone and cold front (Gui et al., 2022; Yin et al., 2022). This dust weather originated from the Gobi Desert area of southern Mongolia and the border between China and Mongolia, and then transported to the southeast along with the northwest airflow, which successively affected North China, Northeast China and Northwest China (Sun et al., 2022; Hu et al., 2023). Due to the characteristics of high intensity, long duration and large impact area, this dust weather has received



widespread attention. So far, many studies have used different satellite remote sensing products to reveal the spatiotemporal
changes, transport processes, pollution levels and radiative forcing effects of this dust weather process (Sun et al., 2022; Hu
et al., 2023; Gui et al., 2022; Liang et al., 2022; Filonchyk, 2022). However, the accuracy, stability, and reliability of these
satellite remote sensing retrieval products are not clear for dust weather monitoring. Furthermore, it is still challenging
whether these data can effectively and accurately capture the dynamic process of dust weather.

Therefore, this study evaluated the accuracy, stability, and reliability of four main satellite remote sensing products (MODIS,
Himawari-8 and Sentinel-5P aerosol products and FY-4A DSD products) that were widely used in dust weather monitoring
studies in terms of the characteristics of the dust event in East Asia in mid-March 2021. By evaluating the accuracy, stability
and reliability of satellite remote sensing in monitoring dust weather in East Asia, it will help us to monitor and study
disaster weather more effectively by using satellite remote sensing technology, and provide reliable scientific basis for
environmental management and danger warning, so as to better maintain human health and ecological balance. The
remainder of this paper is organized as follows: Section 2 introduces the study area, observation data and methods. Section 3
evaluates the continuity, accuracy, and stability of different satellite remote sensing products for dust weather monitoring in
East Asia, and discusses the possible reasons for the discrepancy between satellite remote sensing results and observed
values. Finally, conclusions are given in Section 4.

## 2 Materials and methods

### 2.1 Study area and observation data from ground stations

Spring (March-May) is the season when dust weather occurs frequently in East Asia. On the one hand, the frequent activity
of cold air in northern East Asia in spring provides a driving force for the formation of dust weather. On the other hand,
western East Asia is located inside the Eurasian continent, which is far away from the ocean and has little precipitation. It is
mainly arid and semi-arid areas, with desert areas widely distributed, which provides sufficient material basis for the
formation of dust weather. When large-scale dust weather occurs, the dust content in the air will increase significantly,
exacerbating air pollution. The PM10 concentration data from ground observation stations can effectively represent the dust
content of the air. It is an important data source to reflect the intensity of dust weather. In this study, hourly PM10
concentration data from 925 national environmental monitoring stations in China were selected to evaluate the continuity,
accuracy and stability of dust monitoring activities in East Asia by different satellite products based on the influence range of
dust weather processes from March 13 to 20, 2021 (Hourly PM10 concentration data were obtained from China National
Environmental Monitoring Station: http://www.cnemc.cn/). The location of the study area and the distribution of PM10
monitoring stations are shown in Fig. 1. The background of Fig. 1 is the FY-4A AGRI false-color composite image at 5:00
on March 15, 2021 (UTC), showing the transmission of dust from northwest China to the southeast, and the affected area
extends from northwest China to north and east China coastal areas.



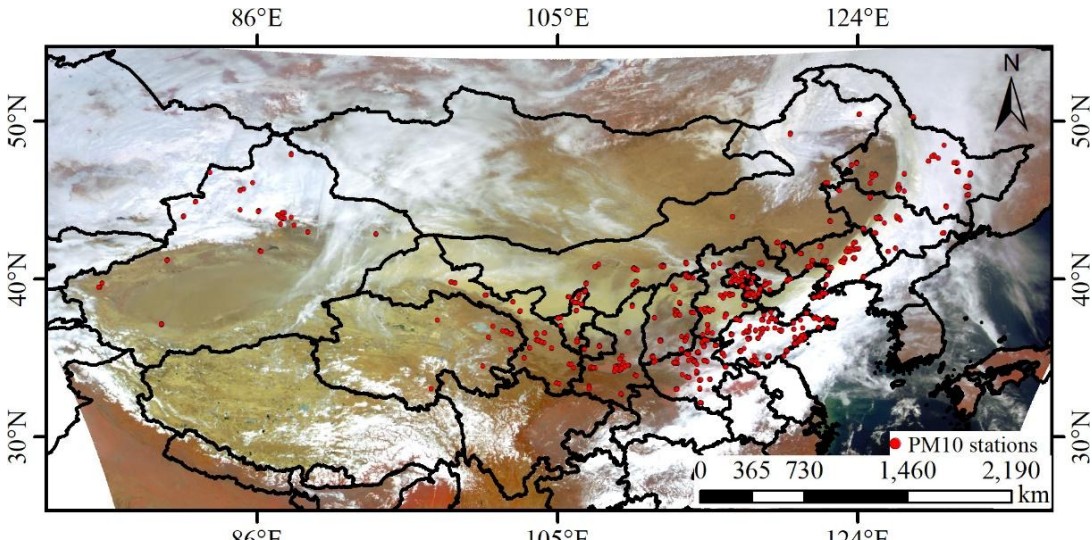

**Figure 1: Location of the study area and distribution of environmental monitoring stations in China**

## 2.2 Observation data from satellites

### 2.2.1 FY−4A

FY-4A is the first satellite of China's new generation of geostationary meteorological satellites. It was launched on December 17, 2016. It is equipped with an advanced geostationary radiation imager (AGRI), geostationary interferometric infrared sounder (GIIRS), lightning mapping imager (LMI) and space environment monitoring instrument package (SEP) can provide continuous monitoring data for land surface, water body, lightning and space weather. Among them, AGRI is the core instrument of FY-4A, which can obtain high spatiotemporal resolution images in 14 spectral bands from visible light to infrared light, with spatial resolution from 0.5 to 4km.

FY-4A not only provides level1 data of original observation, but also provides 32 level2 quantitative satellite observation products, including cloud and atmospheric products, surface products, weather products and radiation products. The data are publicly available from the National Satellite Meteorological Center (NSMC) of China (https://www.nsmc.org.cn). The DSD observed by AGRI was used in the study, with a spatial resolution of 4km and a temporal resolution of 15 minutes. DSD contains DST and IDDI data. Among them, DST is based on the radiation characteristics of dust and uses the characteristics of the AGRI spectral channel characteristics to summarize and generalize the bands and indicators that can be applied to satellite remote sensing dust monitoring, and designs 12 sets of dust identification indicators. Based on dust weather occurrence areas, cloud areas, vegetation areas and desert areas, different reliability indicators are set for the threshold ranges of different identification indicators by using the probability density function (PDF) and cumulative probability density function (CDF) (Zhang et al., 2019). The final dust detection product is synthesized by the PDF of the reliability of 12 identification indicators and their threshold, which effectively avoids misjudgments and omissions caused by a single or a



few identification indicators. The higher the DST, the closer it is to the standard dust spectrum statistical value, that is, the greater the possibility of dust. The recommended DST is above 16 for dust pixels, between 14 and 16 for possible dust pixels, and below 14 for non-dust pixels. Therefore, the study considered that DST greater than 16 is dust.

IDDI describes the difference between the real-time target brightness temperature observed by satellites and the surface brightness temperature of the clear sky atmosphere at the same moment to obtain the attenuation of the brightness temperature of the earth-atmosphere system caused by atmospheric dust aerosols, which can semi-quantitatively characterize the dust intensity (Hu et al., 2007; Legrand et al., 2001). The first step in the generation process of IDDI is to establish a clear-sky surface background brightness temperature image, which consists of the daily maximum thermal infrared brightness temperature value of the surface collected in the latest period. In order to make each image pixel clear-sky observation data and avoid the influence of seasonal changes, it is appropriate to use 10 consecutive days as the synthesis period of the background brightness and temperature image. Secondly, the background brightness temperature image is used to subtract the daily real-time observed brightness temperature image to obtain the brightness temperature difference image, which represents the brightness temperature attenuation caused by atmospheric components (aerosol, water vapour or clouds). Finally, the cloud coverage area is filtered out to eliminate the influence of background aerosols and water vapour. The IDDI value of the remaining area is the IDDI value of atmospheric dust aerosol. Generally speaking, the amount of surface brightness temperature attenuation caused by atmospheric background aerosols and clear-air atmospheric water vapour in sandstorm weather is a small amount compared to the brightness temperature attenuation of dust aerosols, and can be ignored for the identification of dust. Therefore, the image obtained after removing the cloud area can be used as an IDDI image. Generally, the higher IDDI value indicates that the dust content of the air is higher, otherwise it is lower.

In addition, the wavelengths of the first three channels of AGRI are 0.47 μm, 0.65 μm and 0.83 μm respectively, which roughly correspond to the blue (B), green (G) and red (R) channels of visible light. However, its spectral wavelength is not in the optimal range of the three color channels, and the RGB composite image deviates greatly from the true color image. As shown in Fig. 1, the composite image is reddish, and the distinction between typical vegetation areas and bare soil areas is not obvious, but it can better show the dust areas in the atmosphere. Therefore, the study uses the first three bands of AGRI to generate false color composite images to better determine the areas affected by dust.

### 2.2.2 MODIS

Modis carried on the Terra and Aqua satellites is a key instrument in the United States earth observation system program for observing global biological and physical processes. It has 36 medium-resolution spectral bands (0.4-14.4μm). The double satellites combine to observe the earth's surface every 1-2 days to achieve long-term observation of changes in land, oceans, atmosphere and other targets. Its observational data and products are widely used in regional/global ecological environment and natural disaster monitoring and climate change research. The data can be downloaded for free from MODIS Web (https://modis.gsfc.nasa.gov/).



Modis operationally offers two kinds of aerosol products with long-term and global coverage. One is an atmospheric aerosol daily product with a spatial resolution of 10km and 3km, and the other is an atmospheric aerosol daily, 8-day and monthly

composite product with a spatial resolution of 1° × 1°. These aerosol products are based on two famous aerosol retrieval algorithms, including the Dark Target (DT) algorithm on land/ocean and the Deep Blue (DB) algorithm on land (Hsu et al., 2013; Levy et al., 2013). DT algorithm was developed to retrieve aerosols on the surfaces of dark target, while the DB algorithm was mainly designed to overcome the poor retrieve results of the DT algorithm in areas with high reflectance. In order to improve the coverage of data, based on the independent MODIS monthly normalized difference vegetation index

(NDVI) data and taking advantage of the advantages of DT and DB algorithms, an AOD data set combining DT and DB algorithms (DTB) was introduced on land (Sayer et al., 2014). DTB is created as follows: 1) If NDVI > 0.3, use the DT algorithm for retrieval; 2) If NDVI<0.2, use the DB algorithm for retrieval; 3) If 0.2 ≤ NDVI ≤ 0.3, use the average of the DT and DB algorithm retrieval value or anyone through high-quality assurance (QA) control (QA=3 for DT, QA≥2 for DB). Recently, the Collection 6.1 aerosol products were released based on significant improvements in radiation correction and

aerosol retrieval algorithms, with many improvements over the previous Collection 6.0 (Wei et al., 2019). The main improvements of the DT algorithm include 1) the urban surface reflectance model was revised based on MODIS surface reflectance products; 2) in the 10×10km land grid, if the coast or water body pixels exceed 50% and 20%, then QA=0. The main updates of the DB algorithm include 1) heavy smoke detection to solve the problem of over-screening while minimizing the impact of clouds; 2) reduction of artifacts on heterogeneous terrain; 3) improved surface modelling of

elevated terrain; 4) updated regional and seasonal aerosol models over land. However, the method of combining DTB datasets has not changed in any way. The MOD04_L2 and MYD04_L2 aerosol products of MODIS Collection 6.1 version are mainly used in the study, with a spatial resolution of 10km and a temporal resolution of 5 minutes.

### 2.2.3 Sentinel-5P

Sentinel-5 Precursor (Sentinel-5P) is a solar synchronous orbit satellite launched by the European Space Agency (ESA) in

October 2017. It observes the earth at the same angle as the sun and can achieve daily global coverage. The TROPOMI is its key payload, covering all bands from ultraviolet to shortwave infrared. The scanning swath width is about 2600km, and the sub-satellite point spatial resolution is 7×3.5km. It can effectively observe trace gas components in the atmosphere around the world, including NO2, O3, SO2, HCHO, CH4 and CO and other important indicators closely related to human activities, and strengthen the observation of aerosols and clouds (Veefkind et al, 2012). It is currently the imaging spectrometer with

the highest spatial resolution and the most advanced technology in the world that can be used for atmospheric environment monitoring. Its performance has been greatly improved compared to the on-orbit OMI (Fioletov et al., 2020).

The TROPOMI L2_AER_AI dataset provides global high-resolution images of the ultraviolet aerosol index (UAVI), also known as the absorption aerosol index (AAI). UVAI is based on the wavelength-dependent variation of Rayleigh scattering within the UV spectral range for a given wavelength pair, calculating a ratio from the measured top of the atmosphere (TOA)

reflectance and the pre-calculated theoretical reflectance of atmospheric Rayleigh scattering (Apituley et al., 2022). However,



the difference between the observed and simulated reflectance produces residual values. When this residual value is positive, it indicates the presence of UV-absorbing aerosols such as dust and smoke (Michailidis et al., 2023). Therefore, AAI can effectively track the evolution of intermittent aerosol plumes caused by dust weather, volcanic eruptions and biomass burning. Clouds produce residual values close to 0, and strongly negative residual values can indicate the presence of non-

absorbing aerosols and clouds. AAI is a reliable calculation method that depends on aerosol layer characteristics, including aerosol optical thickness, aerosol single scattering albedo, aerosol layer height and underlying surface albedo (Torres et al., 2020). It is well-documented based on years of data, and its key advantages include fast computing speed, wide global coverage, ease of use, and the potential to build long-term climate data records (Apituley et al., 2022). Additionally, unlike satellite-based AOD measurements, AAI can be calculated in the presence of clouds, enabling daily global coverage. In the

study, the AAI index used was measured at wavelengths of 354nm and 388nm, which are wavelengths with very low ozone absorption. According to the study results of Rezaei et al. (2019) and Filonchyk et al. (2020), the study considers that when the AAI>0.7, it is a dust pixel. In order to improve data quality and eliminate the impact of sunlight flicker, only TROPOMI pixels with QA greater than 0.8 are used according to official recommendations. Data are freely and publicly available from: https://dataspace.copernicus.eu/.

**2.2.4 Himawari-8**

In October 2014, the Himawari-8 geostationary meteorological satellite developed by the Japan Meteorological Agency (JAM) was successfully launched, which can achieve regional (80°E-160°W, 60°S-60°N) high-frequency observation, with a maximum spatial resolution of 500m and a temporal resolution of 10 minutes (six full-disk images per hour). Compared with the previous Himawari-7 satellite, the Himawari-8 satellite has been greatly improved in terms of service life and

meteorological observation capabilities. It carries an AHI with 16 bands, including three visible channels and three near-infrared and ten infrared channels. Due to its advantages of high-frequency imaging, high spatial resolution and wide spectral band, it can observe targets such as land, ocean and atmosphere more accurately and detailed (Bessho et al., 2016; Wei et al., 2019). Therefore, it is widely used in monitoring research on clouds, aerosols, sea surface temperatures and natural disasters. In addition, since its spectral band contains aerosol-sensitive blue light channels, it has great potential in

aerosol retrieval (Ge et al., 2018).

Currently, the Earth Observation Research Center of Japan Aerospace Exploration Agency (JAXA) provides level 2 and level 3 aerosol datasets, including full-disk AOD and Ångström index (AE) at 500m wavelength. The AHI aerosol products are generated by an aerosol retrieval algorithm developed by Fukuda et al. (2013). The algorithm is based on the Lambertian assumption on land and sea, using 3 visible bands (470nm, 510nm and 640nm) and 2 near-infrared bands (860nm and

1600nm), introducing weights into the objective function of each channel (Fukuda et al., 2013; Yoshida et al., 2018). Then, the best channel for aerosol retrieval is automatically selected. The specific process is as follows:

Firstly, the radiation correction of clear sky pixels is carried out by assuming that the atmospheric scattering type is only Rayleigh scattering. Then, the pixel with the second lowest reflectance at 470nm within a month is synthesized. It is



considered that the value at 470nm is higher than the value at 640nm due to the influence of residual aerosol pollution. Therefore, using the spectral dependence of surface reflectance, they will be replaced by reflectance calculated as a function of the vegetation index, and these results will be regarded as the true surface reflectance (Zhang et al., 2019). Next, the atmospheric radiation transmission system is used to simulate the reflectance of the top of the atmosphere, and the calculation speed is accelerated by building a lookup table (Wei et al., 2019; Zhang et al., 2019). Based on cluster analysis of AERONET measurement data, the aerosol model was assumed to be composed of fine-particle aerosols (including rural, sea-spray, and yellow dust) and coarse-particle aerosols (including pure ocean and dust) with single-peak lognormal volume distribution (Wei et al., 2019). The cloud contamination was minimized during aerosol retrieval, which was based on a universal cloud detection algorithm previously developed by Yoshida et al. (2018). The empirical approximation method based on MODIS uses the total amount of ozone and water vapor column in OMI and JMA global analysis data to correct the absorption of ozone and water vapor (Wei et al., 2019). Finally, the objective function is established using the simulated and observed TOA reflectance, and those parameters that minimize the objective function are the retrieval results (Zhang et al., 2019). The study used AOD and AE datasets from AHI level 2 hourly (UTC 0:00 to 12:00) and level 3 daily synthetic aerosol products to evaluate the continuity, accuracy and stability of AHI aerosol products in monitoring dust activities in East Asia. Among them, level 3 data is based on the hourly combination algorithm developed by Kikuchi et al. (2018), which is more accurate. The data are publicly available from: https://www.eorc.jaxa.jp/ptree/index.html.

## 2.3 Methods

In this study, the method of AOD and AE (Ångström exponent) relationship was used for aerosol type identification. This method has been used in several studies and its principle is based on the sensitivity of two wavelength parameters to various microphysical properties of aerosols (Boiyo et al.,2018;). AOD is a key physical quantity that characterizes the degree of atmospheric turbidity. It describes the light attenuation caused by aerosol absorption and scattering and its size mainly depends on the aerosol column density. AE is the main indicator to characterize the size of atmospheric aerosol particles, which describes the dependence between AOD and wavelength. When the AE value is less than 1, it indicates that coarse particle aerosols dominate. Otherwise, fine-particle aerosols dominate. Therefore, the dominant aerosol types in the atmosphere of this region can be identified based on the interaction between AOD and AE. In the study, MODIS AOD pixels with AOD > 0.6 and AE < 0.7 are considered to be dust based on the study results of Filonchyk et al. (2020). Similarly, according to the study results of Sun et al. (2022), AHI AOD pixels with AOD > 1.2 and AE < 0.8 are considered to be dust. The continuity, accuracy and stability of satellite remote sensing detection of dust in the atmosphere require verification of ground data. At present, most studies use aerosol parameters measured by AERONET to verify the results of satellite remote sensing detection (Wei et al., 2020). However, the number of AERONET stations in East Asian dust source areas and transmission paths is small and unevenly distributed, making it difficult to provide effective observation data. However, because ground environmental monitoring stations have the characteristics of high observation frequency, large number of stations, and wide distribution, they are often used to verify the authenticity of satellite remote sensing products. In addition,



the PM10 concentration at ground environmental monitoring stations is very sensitive to changes in the concentration of coarse particles, especially changes in dust concentration (Capraz and Deniz, 2021). Therefore, PM10 concentration data from ground monitoring stations provides a reliable and stable data source for verifying the continuity, accuracy and stability

of satellite remote sensing detection of atmospheric dust. According to the "Technical Regulations on Classification of Dust Weather" issued by the Ministry of Ecology and Environment of the People's Republic of China, an hourly PM10 concentration greater than 600 μg·m-3 is considered to be dust weather (Yang et al., 2023). Therefore, when the PM10 concentration at any environmental monitoring station was greater than 600 μg·m-3 during dust weather, which is classified as dust observation. In the study, the probability of correct detection (POCD) and the probability of false detection (POFD)

were used to evaluate the accuracy of satellite remote sensing in detecting atmospheric dust. The calculation formula is as follows:

$$POCD = \frac{DD}{DD+DN}, \tag{1}$$

$$POFD = \frac{ND}{DD+ND}, \tag{2}$$

where, DD represents the number of matching points with PM10 concentration greater than 600μg·m-3, and the satellite

remote sensing detection result is dust. DN represents the number of matching points with the PM10 concentration greater than 600μg·m-3, while the satellite remote sensing monitoring results indicate no dust. ND represents the number of matching points with the PM10 concentration lower than 600μg·m-3, while the satellite remote sensing monitoring results are dust.

In addition, due to the differences in observation frequencies and observation ranges of different types of satellites. In order

to better compare the continuity, accuracy and stability of different satellite remote sensing products in monitoring the evolution of dust weather in East Asia. The study believes that the MODIS, AHI and AGRI data within 15 minutes before and after TROPOMI imaging could all match each other, and the maximum difference in observation time between different satellites did not exceed 30 minutes.

## 3 Results and discussions

### 3.1 Performance of FY-4A in monitoring dust weather

Satellite remote sensing monitoring of dust weather not only provides a wider field of view than ground monitoring stations, but also provides a more intuitive dynamic evolution process of dust weather. Figure 3 shows the spatial distribution of FY4A daily average DST and IDDI during the dust weather process from March 13 to 20, 2021. It can be seen from the figure that there was small-scale dust weather in northwest China, southwestern Inner Mongolia and northwestern Mongolia,

and the intensity was low. On the 14th, dust weather began to intensify in western Mongolia. At this time, the DST in the area affected by dust weather was greater than 19, and the IDDI was greater than 20. At the same time, the dust weather in



southwestern Inner Mongolia also showed an increasing trend compared with the 13th. By the 15th, the scope of the dust weather had formed a dust belt from west to east in northern China, with the affected area extending from eastern Xinjiang to northeastern China. The DST of the entire dust belt was greater than 19, the IDDI ranged from 10 to 30, and the IDDI in the west was larger than that in the east, indicating that the intensity of dust weather in the west was more severe than in the east. In addition, the dust weather activities in the Tarim Basin and northwest Qinghai increased sharply on that day compared with the previous two days. From the 16th and 17th, the area affected by dust shrank from east to west, and the areas with higher intensity of dust weather appeared in southern Mongolia and western Inner Mongolia. The southern part of North China and the northern part of East China were still affected by this strong dust weather process, but the intensity had decreased by March 17. However, the dust weather in the Tarim Basin had become more active, and its scope of influence had rapidly expanded to the east. From the 18th to the 20th, the intensity of the dust weather that occurred in northern China and southern Mongolia gradually weakened. The areas affected by the dust weather shrank to the west, mainly in northwest China and western Mongolia. Affected by the remnants of this dust weather, the dust intensity in North China on the 20th was higher than that on the 19th.

As the main component of dust aerosols, the PM10 concentration at ground monitoring stations is often used to characterize the intensity of dust weather and the degree of pollution caused by the dust weather process. Therefore, PM10 concentration data from 925 national Environmental monitoring stations in China were selected for further analysis in order to obtain more detailed information on the strong dust weather process that occurred in East Asia from March 13 to 20, 2021. The daily average PM10 concentration calculated based on the hourly PM10 concentration of the ground station is shown in Fig. 3. It can be seen from Fig. 2 that North China was the first region affected by dust weather, followed by the Northeast region, and finally the Northwest region. On March 13, the area with the largest daily average PM10 concentration was in southwestern Inner Mongolia, within the range of 500-600 μg/m3. On the 14th, the daily average concentration of PM10 at some stations in this area exceeded 600 μg/m3, indicating that dust weather began to occur in the area with weak intensity. As of the 15th, the average daily concentration of PM10 in more than 90% of stations in North China exceeded 600 μg/m3, and the area has been covered by strong dust weather. Affected by dust transmission, the PM10 concentration at most stations in the Northeast increased significantly on the 15th compared with the 14th, with the average daily concentration being 200-400 μg/m3, and the PM10 concentration at some stations exceeded 500 μg/m3. The dust weather continued to develop on the 16th, and the affected areas expanded to southern and western China. However, the average daily concentration of PM10 in North China and Northeast China showed a downward trend. From March 17th to 19th, the coverage of this dust weather began to shrink from east to west, and the PM10 concentration in North China and Northeast China gradually decreased. Stations with PM10 concentration greater than 600 μg/m3 mainly occurred in the southwest and northwest of Inner Mongolia. There was a significant reduction trend compared with the 16th, and the dust weather was generally ending. After the 19th, affected by the remnants of this dust weather and the easterly airflow, the daily average concentration of PM10 in the eastern northwest and northern China increased.





**Figure 2: Spatial distribution of daily FY-4A DST and IDDI during the dust weather process from March 13 to 20, 2021.**







**Figure 3: Daily average concentration distribution of PM10 in areas affected by the dust weather process from March 13 to 20, 2021.**





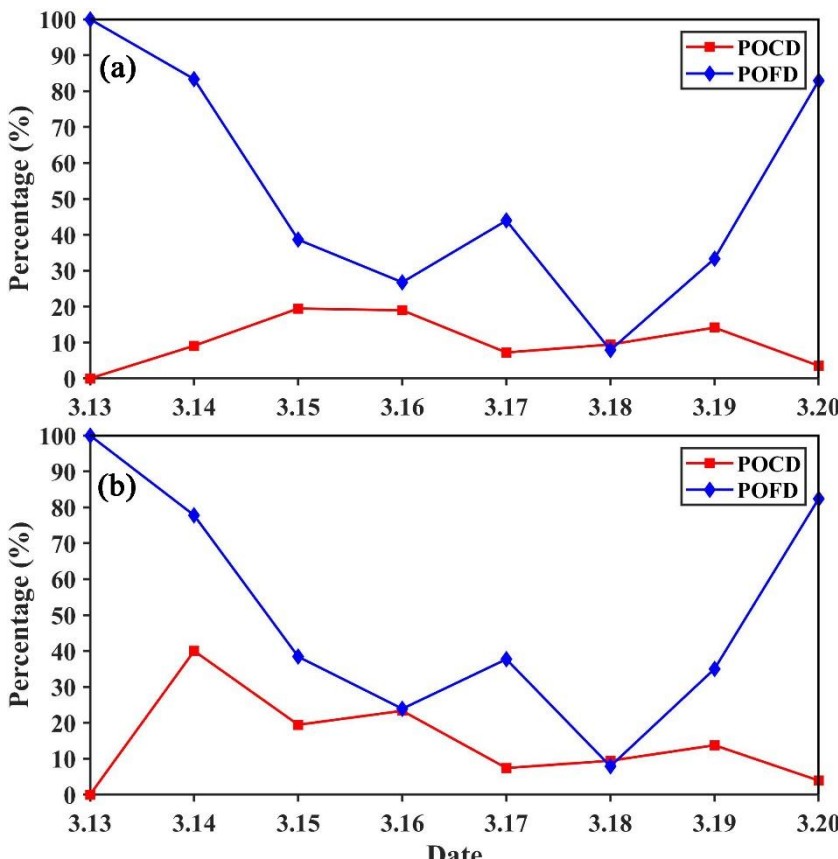

**Figure 4: The POCD and POFD of daily atmospheric dust by FY-4A DST (a) and IDDI (b) product during the dust weather process from March 13 to 20, 2021**

By comparing Fig. 2 and Fig. 3, it can be seen that the changes in the dust weather process captured by FY-4A were in good
agreement with the monitoring results of ground stations, regardless of the intensity or influence range of dust weather. In
order to better prove the application capabilities of the FY-4A DSD product in monitoring dust weather, the study used
PM10 concentration data from ground monitoring stations to further evaluate the atmospheric dust detection capabilities of
the FY-4A DSD product. Figure 4 shows the POCD and POFD of FY-4A DST and IDDI daily atmospheric dust during the
dust weather process from March 13 to 20, 2021. As can be seen from Fig. 4, the POCD of DST dust showed an upward
trend from March 13 to 15, and then showed an overall downward trend, with a maximum value of 19.5% on the 15th.
However, the POCD of IDDI dust reached a maximum of 40% on March 14, and the trend of the POCD from the 16th to the
20th was consistent with DST. The POFD change trend of DST and IDDI was basically the same, showing an overall
downward trend from the 13th to the 18th, and then an upward trend. The minimum POFD of both occurred on March 18,
both of which were less than 10%. In addition, during the whole dust weather process, the dust POCD of DST and IDDI was
overall lower than POFD, and the POFD in the early and late stages of the dust weather were higher than that in the middle
stage, while POCD was opposite.



In addition, from the spatial distribution of daily DST and IDDI in Fig. 2, when the DST judgment result was dust, the IDDI result may not be dust, which was more obvious over desert areas. At the same time, there were also areas where the PM10 concentration at ground stations indicated higher dust intensity, but the IDDI values were smaller. Due to limitations of

observation frequency, cloud pollution, retrieval algorithms and other factors, misjudgment and omission of atmospheric dust detection by satellite remote sensing are inevitable. Although IDDI can be used as a semi-quantitative parameter of dust intensity, it cannot be ignored due to the effect by surface background brightness and meteorological factors (Hu et al., 2007). First of all, the premise of IDDI calculation is that the surface temperature does not change within a certain period of time. However, the surface background brightness temperature changes significantly throughout the day. For East Asia, spring is

not only the season with the most frequent dust weather, but also the season with the most frequent cold air activities. Since the sudden arrival of cold air causes a substantial drop in surface temperature, the synthetic background brightness temperature used in the reference period may be higher than the actual surface brightness temperature on that day, which will cause an overestimation of IDDI. Secondly, during the calculation process, the atmosphere was considered to be clean on the day with the maximum brightness temperature, but the actual situation was not that ideal. In the process of surface brightness

temperature image generation, if there is residual dust in the clearest weather, the final IDDI value will be underestimated. Finally, the maximum brightness temperature in the cloud coverage area that lasts for a long time is still not the actual surface brightness temperature, or the retrieval fog appears when the brightness temperature is synthesized in the reference period, and the retrieval fog is regarded as the surface brightness temperature. Therefore, there will inevitably be systematic errors when calculating IDDI, resulting in the final IDDI value being too high or too low. In addition, Hu et al. (2007) found

that it is difficult to distinguish a small amount of clouds and dust, and the usual cloud detection algorithms often misjudge dust as clouds. Some thin cirrus clouds and heavy precipitation clouds will appear bright temperature difference signals similar to, which is also an important reason for missed and misjudgment of atmospheric dust detected by satellite remote sensing. Duan et al. (2014) compared the forecast results of the FY-2D IDDI product with the GRAPES-SDM dust model and found that the FY-2D IDDI product often misjudged the deep dust in the southern Tarim Basin as a cloud area, which

ultimately Leading to missed judgments of dust. Zhang et al. (2019) used ground dust observation data to verify the IDDI product of FY-4A. The results showed that FY-4A IDDI can detect 88% of ground dust observations. Especially under cloud-free conditions, IDDI can effectively detect dust areas. However, when the dust layer is under clouds or mixed with clouds, the false detection rate is higher.

Although DST is synthesized using the reliability of 12 kinds of dust discriminant indicators, it can avoid misjudgments and

omissions caused by one or a small number of concentrated discriminant indicators to a certain extent. However, these judgment indicators include the 11μm background brightness temperature and actual brightness temperature difference. Therefore, the same error sources as IDDI inevitably occur (affected by surface background brightness temperature and meteorological factors), resulting in misjudgments and missed judgments in the final identification results. Zhang et al. (2019) conducted a preliminary verification of the FY-4A DST product using surface weather phenomena and visibility

observation data. The results showed that the FY-4A dust detection algorithm can effectively identify dust weather processes,



especially for cloudless dust weather detection rates are high. However, there was a certain missed detection in the case of dust mixed with clouds or under the cloud. In addition, an index is usually calculated using the brightness temperatures of two or more thermal infrared bands in these methods, and a fixed threshold is also used to identify dust. However, the brightness temperature observed by satellites is directly related to surface temperature and emissions, and is also affected by

dust characteristics (particle size and vertical distribution, etc.) (Li et al., 2021). Therefore, there are large differences in dust identification results based on a single fixed threshold, and there are significant differences in dust events in different regions and different intensities.

## 3.2 Performance of MODIS in monitoring dust weather

Generally speaking, when large-scale dust weather occurs, the main pollutant in the atmosphere is dust. Therefore, AOD

mainly comes from the contribution of dust aerosols. Figure 5 shows the daily atmospheric AOD spatiotemporal characteristics of the dust weather process retrieved by the MODIS DT algorithm, the DB algorithm and the DTB algorithm. It can be seen that the spatial distribution of daily AOD in dust weather processes retrieved by the DB and DTB algorithms was highly consistent with the spatiotemporal distribution of FY-4A DSD and PM10 concentration by comparing Fig. 5 with Fig. 2 and Fig. 3. In particular, the regions with high AOD values corresponded to the high value area of FY-4A IDDI and

PM10 concentration. However, the spatiotemporal distribution of AOD retrieved by the DT algorithm was not ideal for studying and analyzing the dynamic evolution process of dust weather in East Asia. The main reason is that the high surface reflectance in desert areas has a greater impact on the reflectance of the top of the atmosphere in the red and short-wave infrared bands, resulting in the linear relationship between the surface reflectance in the red and blue bands and the surface reflectance in the short-wave band does not hold, and it is difficult to distinguish the contribution from aerosols and the

ground (Hu et al., 2013). Therefore, compared with the DB algorithm, the DT algorithm is not suitable for AOD retrieval in areas with high surface reflectance. In addition, by comparing the spatial distribution of AOD retrieved by DB and DTB, it was found that DB is significantly better than DTB in describing the details of dust weather. So, the study chosen to further analyze the AOD retrieved by the DB algorithm to evaluate its accuracy and stability in dust weather monitoring.





Figure 5: Spatial distribution of daily MODIS AOD during the dust weather process from March 13 to 20, 2021.



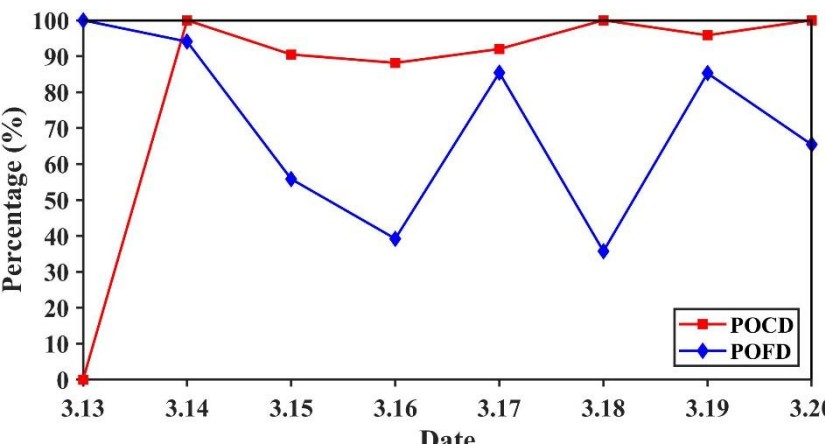

**Figure 6: The POCD and POFD of daily atmospheric by MODIS AOD during the dust weather process from March 13 to 20, 2021.**

Figure 6 shows the daily atmospheric dust POCD and POFD of AOD retrieved by MODIS DB algorithm during the dust weather process from March 13 to 20, 2021. As can be seen from Fig. 6, except for the 13th day, the average accuracy rate of atmospheric dust detection by AOD retrieved by the DB algorithm was higher than 90%, and the overall change was relatively stable. Among them, the POCD on the 14th, 18th and 20th was 100%. It can also be found from Fig. 6 that the POFD change trend of the AOD fluctuated greatly during this dust weather. The POFD in the early stage of dust weather was higher than that in the middle and late stages. The maximum POFD was 100% on the 13th, and the minimum POFD appeared on the 18th, about 36%. One reason for the high POFD is that most PM10 stations are located in eastern and central China, with relatively few in the northwest where dust weather originated. Therefore, low DD values led to high POFD in the early stages of dust weather. Secondly, the PM10 concentration greater than 600µg/m3 was considered to be the occurrence of dust weather, which may lead to excessively high ND, resulting in high POFD. In addition, since dust would settle in the process of diffusion in the atmosphere, when it was transported to downstream areas, its content was significantly lower than that in upstream areas. The PM10 concentration monitored by environmental stations would not be too high, which will lead to an excessive number of ND, thereby increasing POFD to a certain extent. The above reasons that led to the high POFD of atmospheric dust by MODIS AOD product during dust weather processes are also applicable to other satellite products.

The MODIS aerosol retrieval algorithm is a radiation-based physical method that mainly relies on the physical properties and spectral characteristics of atmospheric aerosol particles in the visible and near-infrared bands, which is currently the most mature and widely used quantitative remote sensing method for dust aerosols (Hsu et al.,2013; Yan et al., 2020; Wang et al., 2020). However, physically empirical methods are limited by their reliance on thresholds that may be a function of land cover type, aerosol properties, lighting conditions, scattering angles, etc. (Li et al., 2021). In order to quantitatively evaluate the impact of surface albedo, Zhang et al. (2018) used the 6S radiation transfer model to simulate the relationship between the difference between the satellite observation apparent reflectance and the surface reflectance at the 0.47µm wavelength under different AOD concentration conditions as the surface emissivity changes. The results showed that visible



light can be used to effectively monitor dust aerosols over the ocean. However, for areas with complex surface types, the contribution of surface reflected radiation needs to be considered. When the surface reflectance is higher than a certain value, the apparent reflectance changes little as the aerosol optical thickness increases, and the radiation observed by the satellite mainly comes from the contribution of reflected radiation from the underlying surface. Therefore, how to eliminate the
influence of the underlying surface from satellite observations is the key and difficulty in improving quantitative remote sensing of aerosols. In addition, visible and near-infrared remote sensing cannot penetrate the clouds to detect dust under the clouds. However, dust weather is often mixed with clouds when they occur, and cloud pollution in dust pixels can lead to an increase in AOD (Li et al., 2021). At the same time, in actual situations, the AOD of atmospheric dust aerosols changes continuously in space and the boundaries are blurred during dust weather processes. Therefore, it is unrealistic to
unambiguously classify pixels into dust, cloud, and clear sky.

**3.3 Performance of Sentinel-5P in monitoring of dust weather**

Positive AAI values indicate the presence of absorbing aerosols in the atmosphere, such as dust, smoke and volcanic ash, while negative values indicate the presence of non-absorbing aerosols (sulfates and sea salts) and clouds (Filonchyk et al., 2020; Penning de Vries and Wagner, 2011). Generally speaking, there are few large-scale straw burning and forest fires in
northern China in spring, let alone volcanic eruption. Therefore, when large-scale dust weather occurred and the AAI was positive, it was basically dust aerosols. Figure 7 shows the daily spatial distribution of AAI during the dust weather from March 13 to 20, 2021, which clearly and continuously reflected the transmission process and intensity of atmospheric dust during this dust weather. The daily AAI spatial distribution retrieved by TROPOMI was consistent with the spatial distribution of atmospheric dust obtained from MODIS AOD and FY-4A DSD products, with AAI values ranging from 0 to
4 in most areas. However, compared with MODIS AOD and FY-4A DSD, its spatial continuity was better. For example, on March 15, TROPOMI AAI captured dust plumes over the Bohai Sea and northeastern Inner Mongolia. However, neither MODIS DB nor FY-4A DSD captured these local pollution points caused by dust weather, which was most likely affected by cloud cover. Therefore, the TROPOMI AAI product has the advantage over other passive remote sensing aerosol products of being able to detect dust under cloudy conditions, making up for the omission of dust information in other
passive remote sensing products due to the influence of cloud coverage. In addition, there were also areas where the AAI value exceeded 4 during dust weather, indicating that the atmospheric dust content in these areas was greater than in other areas.

Overall, AAI was able to capture aerosol plumes from desert dust in great detail due to its high spatial resolution and high signal-to-noise ratio. Next, the study used PM10 concentration data from ground monitoring stations to evaluate the
atmospheric dust detection capabilities of TROPOMI AAI, and the results are shown in Fig. 8. It can be seen from Fig. 8 that the atmospheric dust POCD of AAI during the dust weather process from March 13 to 20, 2023 showed an overall upward trend from March 13 to 19, and then decreased, with the maximum value being 54% of that on March 19. However, the changes in POFD fluctuated more than POCD, with the maximum value appearing on March 18, about 80%. In addition, the





daily POCD changes of AAI during dust weather were generally lower than POFD. During the early and middle stages of dust weather (March 13-18), the overall change trends of POCD and POFD were consistent. Interestingly, the growth rate of AAI POFD was much faster than that of POCD when the dust weather swept across the northern part of China on March 15th.

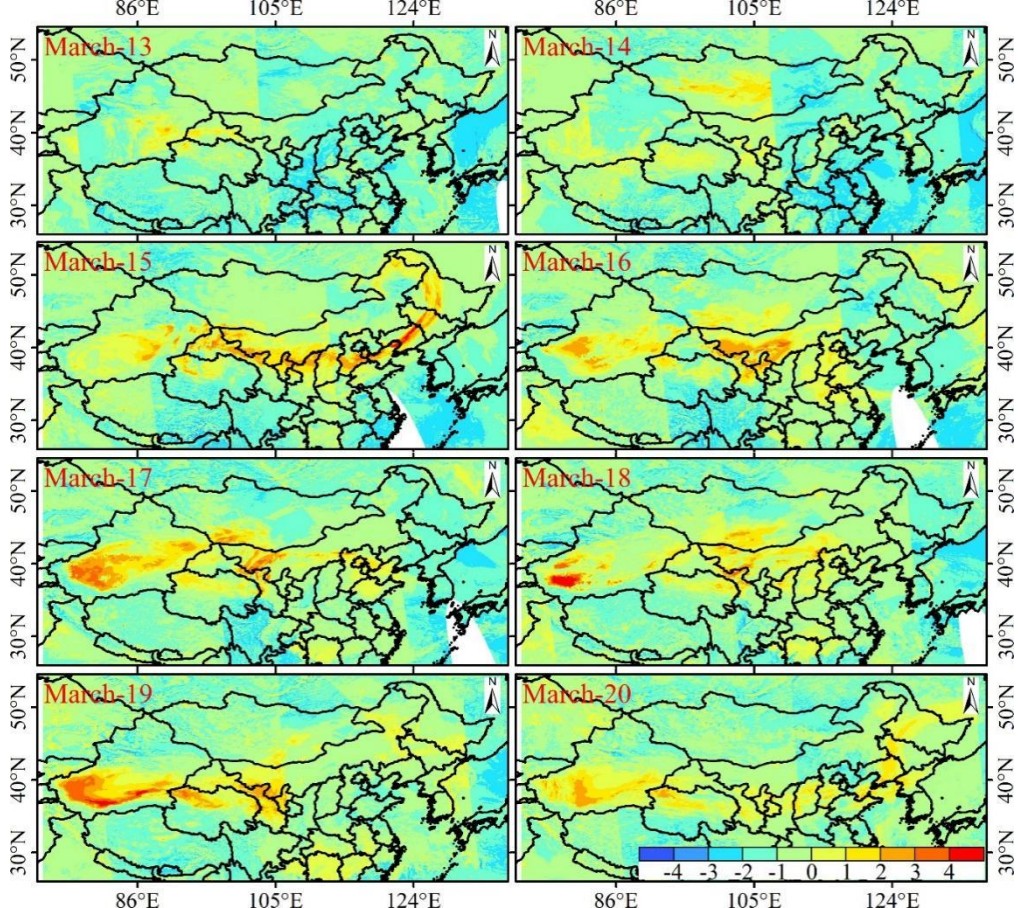

**Figure 7: Spatial distribution of daily Sentinel AAI during the dust weather process from March 13 to 20, 2021.**

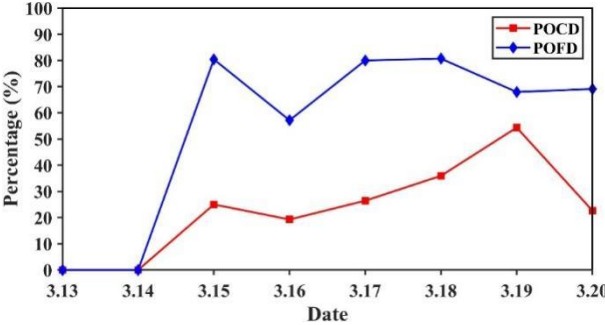


**Figure 8: The POCF and POFD of daily atmospheric dust by Sentinel-5P AAI during the dust weather process from March 13 to 20, 2021.**



As mentioned before, AAI is obtained by calculating the residual value resulting from the ratio between observed and simulated reflectance. Therefore, the AAI calculation method is not a classic aerosol retrieval method. The calculation of AAI relies on measured reflectance, the theoretical atmospheric reflectance with Rayleigh scattering stored in a pre-calculated look-up table, and the assumption that the scene surface behaves as a Lambert equivalent reflector (Dave and Mateer, 1967). As early as 1997, Herman et al. (1997) used the radiation difference between 340 and 380 nm measured by Nimbus-7/TOMS to obtain the global distribution of UV-absorbing aerosols and the interannual variation of aerosols in major desert areas from 1979 to 1993. It was found that atmospheric dust aerosol contributed the most to the absorption of ultraviolet band. However, due to the small amount of Rayleigh scattering in the bottom layer, TOMS was insensitive to the bottom signal. Therefore, it was difficult to obtain information on absorbing aerosols below the atmospheric boundary layer. Apituley et al. (2022) compared the aerosol index data of TROPOMI AAI with OMI and OMPS (Ozone Mapping & Profiler Suite) and found that the values observed by TROPOMI were lower than those of OMI and OMPS. It is known that the accuracy of aerosol products is sensitive to small changes in calibration radiation, which may lead to deviations (Torres et al., 2020; Go et al., 2020). In addition, the deviation of AAI is also dependent on the knowledge of surface albedo and wavelength-dependent variability of surface albedo (Chimot et al., 2017). Some studies have also found that the detection of absorbing aerosols in the UV band and the calculation of optical thickness are affected by the presence of large-scale and sub-pixel clouds in the sensor field of view (Herman et al.,1997; Penning et al., 2011). Moreover, Zweers (2022) used determining instrument specifications and analyzing methods for atmospheric retrieval to test the influence of terrain height variation on AAI. The results showed that when the terrain height is less than 250 m, the AAI deviation is about 0.3 depending on the layer height and layer thickness.

**3.4 Performance of Himawari-8 in monitoring of dust weather**

Figure 9 shows the spatial distribution of Himawari-8 daily average AOD during East Asian dust weather processes from March 13 to 20, 2021. As can be seen from Fig. 9, the spatiotemporal evolution of this dust weather process captured by Himawari-8 daily average AOD was generally consistent with FY-4A DSD, MODIS AOD and Sentinel AAI. At the same time, it can also effectively capture dust transport over the ocean. In this regard, the monitoring of long-distance dust transport in East Asia is superior to the FY4A DSD and MODIS AOD products. In addition, Himawari-8 has a higher spatiotemporal resolution. Therefore, compared with other satellite aerosol products, it has a stronger ability to describe the evolution of dust weather in more detail. However, due to the limitation of the satellite observation range, Himawari-8 cannot effectively monitor dust activities in central and western Xinjiang, China.



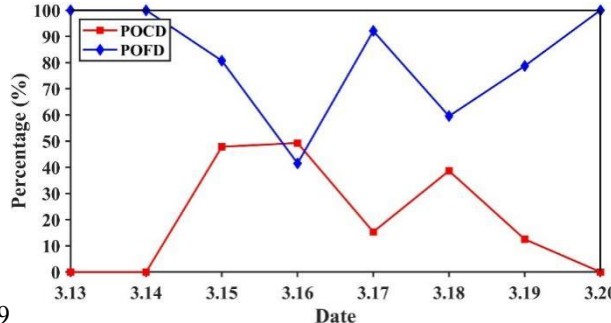

**Figure 9: Spatial distribution of daily Himawari-8 AOD during the dust weather process from March 13 to 20, 2021.**

**Figure 10: The POCD and POFD of daily atmospheric dust by Himawari-8 AOD during the dust weather process from March 13 to 20, 2021.**



Figure 10 shows the POCD and POFD of atmospheric dust by the Himawari-8 AOD product during the dust weather process. It can be seen from the figure that the POCD and POFD of Himawari-8 AOD atmospheric dust showed an opposite trend. After the outbreak of dust weather, the POCD of Himawari-8 AOD atmospheric dust showed a significant upward trend, and the maximum POCD appeared on March 16. At the same time, this was also the day when the product had the smallest

POFD during this dust weather. In general, compared with other satellite aerosol products, the changes in the POCD and POFD of atmospheric dust by Himawari-8 AOD were more consistent with the changes in the overall change process of dust weather (formation, development and demise).

The extremely high observation frequency of Himawari-8 can generate time-continuous aerosol products, thereby effectively obtaining time-continuous daily changes in aerosols. However, small uncertainties in surface reflectance may affect the

accuracy of AOD (Hashimoto and Nakajima, 2017). Tang et al. (2023) studied and analyzed the impact of surface conditions, observation angles and aerosol types on the accuracy of Himawari-8 AOD retrieval. It was found that Himawari-8 AOD has obvious underestimation in areas with surface reflectance close to 0.1 and NDVI close to 0.5, and there is no linear relationship between surface conditions and retrieval accuracy, indicating that Himawari-8 AOD retrieval accuracy does not entirely depend on the surface reflectance. In addition, the study also found that significant underestimation occurs when the

aerosol load is high, coarse particles dominate, and the observation zenith angle is less than 50°. Similarly, Wei et al. (2019) tested the accuracy of Himawari-8 aerosol products using AERONET and Sun-Sky Radiometer observation data from 98 ground stations in the main observation area of Himawari-8, and found that there are large uncertainties in both AOD and AE. AOD captured daily changes well, but performed worst in spring. AE generally showed significant underestimation, especially in China. At the same time, the AOD retrieval accuracy increased with the increase of NDVI and AE, indicating

that the current Himawari-8 aerosol retrieval algorithm was not suitable for the retrieval of atmospheric aerosol optical parameters under bright surfaces and high load conditions of coarse particles. Jiang et al. (2019) used the AOD measurement results of the AERONET sites to evaluate the AOD accuracy of Himawari-8 and MODIS Deep Blue algorithm retrieval in China, and compared them. It was found that the AOD accuracy retrieved by Himawari-8 greatly depends on the atmospheric aerosol load, AE and NDVI. However, the MODIS AOD retrieval bias does not appear to be related to these

variables. Due to the above factors, there are errors in the retrieved AOD, which eventually spread to the actual application of the aerosol product.

### 3.5 Evaluation of difference satellites in monitoring dust weather

In this study, spatiotemporal matching of different satellites was performed to better compare the continuity, accuracy and stability of FY4A DSD, MODIS AOD, Sentinel-5P AAI and Himawari-8 AOD for monitoring the evolution of dust weather

processes in East Asia. Figure 11 shows the spatial distribution of atmospheric dust obtained by FY4A, MODIS, Sentinel 5P and Himawari-8 when the imaging time is closest and there were overlapping coverage areas during the dust weather process. It can be seen from Fig. 11 that due to the different scanning widths of different satellites, there were significant differences in the spatial distribution of atmospheric dust captured by different satellites at approximately the same time. The





geostationary satellites (FY4A and Himawari-8) have a wide imaging range and high observation frequency, and the spatial

continuity of atmospheric dust acquired by them is better than that of polar-orbiting satellites (MODIS and Sentinel 5P). In the same observation area, the distribution of atmospheric dust presented by FY4A DSD, MODIS AOD and Sentinel-5P AAI products had a good consistency. However, the performance of the Himawari-8 AOD product was worse than that of several other products. This can be affected by factors such as cloud coverage and retrieval algorithms. In addition, because Sentinel 5P AAI has the advantage of detecting dust under clouds, it makes up for the shortcomings of other satellite aerosol

products when there is cloud coverage.

**Figure 11: Comparison of different satellite remote sensing products in monitoring dust weather**

Figure 12 shows the atmospheric dust POCD and POFD of the FY4A, MODIS, Sentinel 5P and Himawari-8 aerosol products throughout the dust weather process (Fig. 12a) and spatiotemporal matching (Fig. 12b). As can be seen from Fig.

12a, the MODIS AOD product performed best during the whole dust weather process, with a POCD of 91%. Followed by Himawari-8 AOD, with a POCD of 35.5%. FY4A DST and IDDI had comparable POCD of 14.6% and 15.8%, respectively.



The POCD of Sentinel 5P AAI was 24.4%, which was lower than the POCD of MODIS and Himawari-8AOD products. However, its POCD was higher than the FY4A DSD product. Therefore, purely from the POCD point of view, the MODIS AOD product was better than other satellite aerosol products when used in dust weather monitoring in East Asia. However, all dust aerosol products also had high POFD. Among them, the Sentinel-5P AAI product had the largest POFD of 72.9%. The second was the POFD of MODIS and Himawari-8 AOD products, both at 63.2%. The POFD of FY4A DST and IDDI products was relatively low, 38.8% and 37.1% respectively. In addition, it can be seen from Fig. 12a that the POFD of several other dust aerosol products was much larger than the POCD except for MODIS AOD.

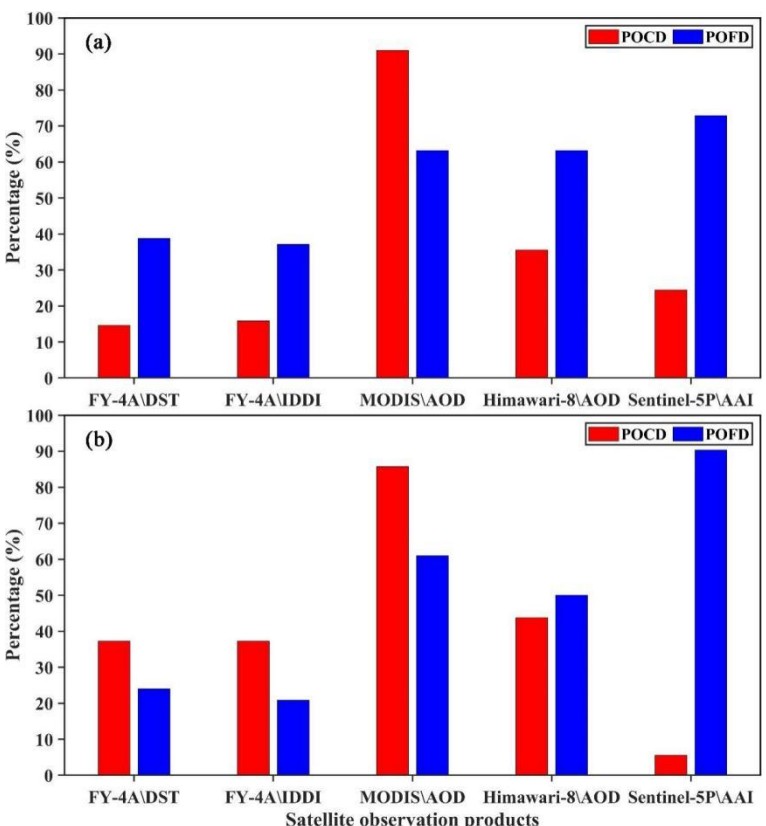

**Figure 12. The POCD and POFD of atmospheric dust by different satellite products under the whole dust weather process (a) and spatiotemporal matching (b) from March 13 to 20, 2021**

It can be seen from Fig. 12b that the POCD of MODIS AOD was still the largest (85.7%) at roughly the same observation time and the same observation area. This was followed by the AOD of Himawari-8 at 43.8%. Next was FY4A DST and IDDI products, both at 37.3%. The smallest POCD was the Sentinel-5P AAI product, which was 5.6%. However, the POFD of the Sentinel-5P AAI product was the largest among all aerosol products at 90.3%. The POFD of the other four products in descending order was MODIS AOD (61%), Himawari-8 AOD (50%), FY4A DST (24%), and FY4A IDDI (20.8%). In addition, the POCD of MODIS AOD and FY4A DSD products was higher than POFD at the same time and space (Fig. 12b).



However, the POCD of Himawari-8 AOD and Sentinel 5P AAI products was lower than POFD. In particular, the POFD of Sentinel 5P AAI was about 85% higher than POCD.

## 4 Conclusions

Satellite remote sensing provides an effective data source for monitoring frequent dust weather in East Asia, which makes up for the shortcomings of conventional monitoring methods. This study uses PM10 concentration data from ground stations to evaluate and compare the continuity, accuracy, and stability of FY4A DSD, MODIS AOD, Sentinel-5P AAI, and Himawari-8 AOD products for dust weather monitoring in East Asia. The main conclusions are summarized as follows:

From the perspective of spatial continuity, the daily atmospheric dust level distribution presented by different satellite remote sensing dust aerosol products had good spatial consistency. In particular, the spatial distribution of atmospheric dust aerosols captured by the Sentinel-5P AAI product. It not only has more details than other products, but also makes up for the deficiency of these satellite remote sensing products in the detection of dust under the cloud. In addition, the high and low distribution of atmospheric dust concentrations captured by these satellites is very consistent for the high and low distribution of PM10 concentrations at ground stations.

Evaluating the atmospheric dust detection performance of different satellite remote sensing products during the whole dust weather process, the MODIS AOD product had the best performance with a POCD of 91%. Followed by the Himawari-8 AOD product, its POCD was 35.5%. Next was the Sentinel-5P AAI product with a POCD of 24.4%. Finally, they were FY4A IDDI and DST products whose POCD was equivalent with 15.8% and 14.6% respectively. In addition, the MODIS AOD product performed best in daily atmospheric dust POFD after the dust weather outbreak, and the changes were the most stable.

From the time and space matching results, the POCD of MODIS AOD products in the same time and space was still the highest, at 85.7%. Next it was the Himawari-8 AOD product with a POCD of 43.8%. Finally, it was FY4A DST and IDDI products with both POFD of 37.3%, and the POFD was smaller than other products. The POCD of Sentinel-5P AAI product was not only the lowest among several products at 5.6%, but its POFD was also the largest at 90.3%.

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
