# Peer review of "Performance and evaluation of remote sensing satellites for monitoring dust weather in East Asia"

_EGUsphere, 2025_

## Author Comment (AC1)

**Response to Reviewer 2**

1. A single dust weather event is insufficient to characterize the accuracy and stability of different products in identifying dust storms. It is recommended that the authors expand the study's temporal scope to include dust event cases across multiple seasons, analyzing the impact of seasonal surface changes, such as vegetation cover, soil moisture, and surface temperature, on the accuracy and stability of dust identification by different remote sensing products, (e.g., the high-dust storm period in spring season versus the low-dust storm period in other seasons), thereby evaluating their cross-seasonal stability and adaptability.

Response:In East Asia, dust storm weather mainly occurs in the spring. Based on your suggestion to increase research on multiple dust storm events, we compiled data on dust storm incidents that occurred from April 2015 to 2019 from the National Climate Center of the China Meteorological Administration: http://ncc-cma.net/cn/. We evaluated the performance of various satellite products for these 64 events.

2. In the manuscript, there is an issue with the citation format, for example, on the line 86 of page 3, "which can distinguish dust from complex atmospheric environments. (Liu et al., 2008).", and line 88 of page 3, aerosol-cloud interaction and climate effects, etc. (Gui et al., 2022; Jia et al., 2018; Wang et al., 2023).", and etc.

Response:Thank you for pointing out the issues. We have carefully revised the references in the text.

3. It is recommended that the authors overlay these satellite remote sensing product data with PM10 concentration data from ground observation stations spatially (e.g., displaying satellite dust distribution and PM10 concentration points simultaneously on an image) to more intuitively demonstrate the consistency between product detection results and ground observations, thereby enhancing the readability of figures and the clarity of result interpretation, specifically between Fig. 2 and Fig. 3.

Response:We adjusted the drawing of the Figures and overlaid the PM10 values with the satellite products (Figure 3/5/7/9/11).

4. It mentioned that "when the DST judgment result was dust, the IDDI result may not be dust, which was more obvious over desert areas" on line 377-378, how to explain this situation, and the results in Fig. 2 on March 16, IDDI exhibited this difference compared to DST over the North China Plain. How can this

36       phenomenon be explained?

37    Response:The imprecise statement has been amended in the article.

38    5. Lines 421-423 state, "it was found that DB is significantly better than DTB in
39       describing the details of dust weather." How can this statement be explained, and
40       it is recommended to provide a detailed explanation?

41    Response:In the latest manuscript, we no longer make a comparison between DT and
42       DTB; we have only selected the DT product.

43    6. How was the POCD for all products in Figure 12(a) calculated from March 13 to
44       20? Was it an average of daily results or another method? Referring to Figure 6,
45       the average POCD for MODIS over the whole dust weather process cannot reach
46       91%.

47    Response:In the latest manuscript, for the 64 DEs, each POCD value represents the
48       total value during each DE process. In the original manuscript, the POCD in
49       Figure 6 was the daily average, while Figure 12 represented the total value for
50       the entire dust storm event. Since the daily values of DD, DN, and ND vary, this
51       caused the daily average in Figure 6 to appear inconsistent with the total POCD.

52

53

---

## Author Comment (AC2)

**Response to Reviewer 1**

1. Abstract: Full name of acronym (FY4A, MODIS, and POCD)

Response: Thanks for your suggestion. We have now spelled out the full names of the acronyms (FY4A, MODIS, and POCD) in the abstract upon their first use (Line 13, Line 14, Line 19).

2. Lines 27-29: "When dust weather occurs, … (Mahowald, 2011)": More references should be added here.

Response: We have expanded the discussion on dust weather impacts by adding key references (Mahowald, 2011; Kok et al., 2023; Zhuang et al., 2001) to better support this point in Line 29.

3. Lines 45-47: "After the 1970s, with the rapid development of various earth observation…": Instead of listing references, I suggest briefly introducing key previous studies closely related to this research and linking them individually to the references. This approach should be applied to many other sections of the manuscript as well.

Response: Thanks. We have expanded the discussion of key studies in the specified section (Lines 45-47) by explicitly linking major advancements to their seminal references. This approach has been consistently applied throughout the manuscript to better contextualize our work within existing literature.

4. Lines 53-77: This paragraph introduces the various sensors (distinguishing between GEO/LEO), algorithms, and the variables provided by each algorithm. I recommend to revise it to discuss their strengths and weaknesses and explain the rationale behind the selection of products in this study. For example, this study uses DT and DB products from three available MODIS aerosol products (DT, DB, and MAIAC). VIIRS continues MODIS observations, but the products are not used here. Although Korean Geostationary Satellites such as GEMS, GOCI, and AMI provide aerosol information, they are not used here. Additionally, references are needed for each aerosol product.

Response: In lines 65-75, the referenced data has been supplemented with relevant citations. Additionally, we discussed the advantages and disadvantages of the product in both the data introduction and the results and discussion sections.

5. Lines 86-87 "Its aerosol classification monitoring and vertical structure are currently the most comprehensive and accurate aerosol product" : On what basis is this defined?

Response: In Line 92-94, We have revised the statement to remove unsubstantiated absolute claims while maintaining emphasis on the product's utility, with supporting references added for verification.

6. Line 94 "The strongest dust weather": On what basis is this defined? There are many instances of dust being transported across the Pacific Ocean.

Response: Following your comment, we have conducted a broader analysis of dust events and consequently removed the absolute descriptor "strongest" from the text.

7. Lines 101-104 "However, the accuracy, stability, and reliability of these satellite remote sensing retrieval products are not clear for dust weather monitoring" : That is not true. Here is a quick example: https://doi.org/10.1002/2015JD024103

Response: Lines 103-104, We have revised the expression to "Research assessing the accuracy, stability, and reliability of these satellite remote sensing retrieval products for dust storm monitoring has been scarce. "

8. Line 116 "Spring (March-May) is the season when dust weather occurs frequently in East Asia": Need reference.

Response: Thanks. As suggested, we have added supporting references to this statement in Line 114.

9. Lines 116-117 "the frequent activity of cold air in northern East Asia in spring provides a driving force for the formation of dust weather" : What does the 'cold air activity' refer to?

Response: The cold air activity here refers to the Mongolian cyclone. We have made relevant modifications in the corresponding sections of the article and added references in Line 116.

10. Figure 1: It is difficult to distinguish the dust scene from the background surface in this image. Why not provide a 'background RGB image' from a pristine day as a reference?

Response: We have updated Figure 1 by adding a reference RGB image from a pristine dust-free day to enhance the contrast and visibility of dust features.

11. 2.2.1 FY-4A: Could you summarize the definitions of the various dust indices in a formula or table to allow for a clearer comparison? Which wavelength is used for deriving IDDI?

Response: The DST data documentation does not explicitly provide the formula, and
        some references only offer an introduction to the data. The formula for IDDI has
        been provided in Lines 165-168.

12. 2.2.2 MODIS: Please revise Modis into MODIS in the section.

Response: Thanks. We have corrected "Modis" to "MODIS" throughout Section 2.2.2
        to maintain consistent acronym formatting.

13. Line 177: remove 'the United State'

Response: We have removed "the United State" as suggested.

14. Line 181: remove 'for free'

Response: We have removed " for free " as suggested.

15. Line 183: MODIS provides multiple aerosol products (DT, DB, MAIAC…) and
        each algorithm provides Lv2 and Lv3 products.

Response: In Lines 183-185, we have revised this sentence "MODIS provides aerosol
        products with varying resolutions (1 km, 3 km, 10 km) for operational use,
        offering long-term and global coverage. One type includes daily atmospheric
        aerosol products with spatial resolutions of 10 km and 3 km, while another type
        features daily, 8-day, and monthly composite products with a spatial resolution of
        $1° × 1°$."

16. Lines 187-188: "…while the DB algorithm was mainly designed to overcome the
        poor retrieve results of the DT algorithms in areas with high reflectance.": The
        DT algorithm is designed based on its theoretical background to target dark soil
        and vegetation surface. The Deep Blue algorithm was developed to overcome
        uncertainties in bright surfaces by utilizing observations from the deep blue
        channel. As a result, both the DT and DB algorithms perform complementary
        roles in global aerosol observations.

Response: In Lines 185-192, we have revised this sentence "These aerosol products
        are based on two famous aerosol retrieval algorithms, including the Dark Target
        (DT) algorithm on land/ocean and the Deep Blue (DB) algorithm on land (Hsu et
        al., 2013; Levy et al., 2013). Due to the significant impact of high-reflectivity
        areas such as deserts and snowfields on the atmospheric top layer reflectance in
        the red light and shortwave infrared bands, the linear relationship between the
        surface reflectance of red and blue light (0.65 and 0.47 μm ) and the surface reflectance in the shortwave infrared band (2.11 μm) does not hold. This makes it difficult to distinguish the contributions from aerosols and the ground (Hsu et al., 2013). In contrast, the DB algorithm shows better retrieval results in these areas, as its initial development aimed to overcome the uncertainties in retrieval results in high-reflectance environments."

17. Line 194: Collection 6.1 is the latest version, though it was not released recently. A new DT GEO-LEO combined products is available here: ladsweb.modaps.eosdis.nasa.gov/missions-and-measurements/applications/geoleo/

Response: Thank you for your reminder. We have revised the description of Collection 6.1 in Line 195.

18. Lines 207-209 "It can effectively observe trace gas components in the atmosphere around the world…": TROPOMI/S5P instrument provides hyperspectral measurements in UV visibie, NIR and SWIR, which are also advantageous for retrieving aerosol absorptivity (like SSA) and aerosol layer height.

Response: In Lines 199-202, we have revised this sentence "As the world's highest-resolution and most advanced imaging spectrometer for atmospheric environmental monitoring, TROPOMI provides hyperspectral measurements across ultraviolet (UV), visible (VIS), near-infrared (NIR), and shortwave infrared (SWIR) bands (Veefkind et al, 2012)."

19. Line 221: aerosol optical thickness aerosol optical depth

Response: We modified this word in Line 213.

20. Line 249-251: The sentence here is unclear. What does "the value at 470 nm is higher than the value at 640 nm" means? "…using the spectral dependence of surface reflectance…": Does it means the surface reflectance relationship suggested in Kaufman et al. (1997)?

Response: Based on your feedback, we have reorganized this sentence in Lines 239-243 as follows: Then, the pixels with the second lowest reflectance at 470nm within a month are synthesized. Pixels exhibiting values at 470 nm that are higher than those at 640 nm are suspected of being influenced by residual aerosol contamination. To address this, these pixels will be replaced with reflectance values calculated based on the vegetation index, utilizing the spectral dependence of surface reflectance (Kaufman et al., 1997). These results will be considered as the true surface reflectance.

21. 2.3 Method: I would recommend summarizing the criteria for strategy for detecting dust pixels in a table or diagram.

Response: Based on your suggestions, we have added Table 1 in Section 2.3 to
summarize the criteria for detecting dust pixels.

22. Line 261 "…its size": 'magnitude' might be better than 'size'.

Response: We have modified this.

23. Line 281 "ground environmental monitoring stations": Authors need to provide
the characteristics of the PM10 observation (ex. Retrieval frequency, accuracy,
sensor calibration…)

Response: The observational characteristics of PM10 have been introduced in Section
2.1.

24. Line 307 "Figure 3": Maybe Figure 2?

Response: Thank you for your reminder. We have carefully checked the details in the
text.

25. Figure 2: Please indicate the region mentioned in the text on the figure. This will
make it easier to follow the discussion.

Response: The mentioned parts have been labeled in Figure 1. For the areas that could
not be marked in Figure 1, we have also included the latitude and longitude
range at the first mention.

26. Figure 2 and Figure 3: I recommend aligning the projection areas to facilitate
comparison between the dust index and $PM_{10}$ concentration. Why not consider
combining Figure 2 and Figure 3?

Response: Following your recommendation, we have merged this and processed
similar Figures within the text.

27. Figure 4: It is recommended that each figure be accompanied by a title. It is
questionable whether analyzing trends in this figure is appropriate, given that
POCD and POFD are unlikely to vary continuously across space and time.

Response: First, we changed the chart from a line graph to a bar chart, and secondly,
we will no longer analyze the trends.

28. Lines 355-358 "In order to better prove the application … the atmospheric dust
detection capabilities of the FY-4A DSD product.": This appears to be an unnecessary statement. Removing it will improve conciseness and flow without losing important content.

Response: Okay, this sentence in the text has been deleted.

29. Lines 370-371 "misjudgment and omission of atmospheric dust detection by satellite remote sensing are inevitable": The current phrasing incorrectly generalizes limitations to all satellite remote sensing. This should be narrowed to specifically address the FY-4A dust products being discussed.

Response: Okay, we have added the qualifier 'of FY-4A/B' at the end.

30. Line 409 "Generally speaking, when large-scale dust weather occurs, the main pollutant in the atmosphere is dust." I guess it depends on season, time, and location.

Response: Thank you for your reminder. We have modified this sentence in Lines 409-410 as follows: For East Asia, especially in northern China, when large-scale sandstorms occur in the spring, the main pollutant in the atmosphere within the affected area is dust (Filonchyk, 2022; Song et al., 2022)

31. Line 420 "Therefore, compared with the DB algorithm, the DT algorithm is not suitable for AOD retrieval in areas with high surface reflectance.": The current statement presents an overly simplistic and potentially misleading view of the algorithms' capabilities. While the DB algorithm performs better in areas with bright surface, the DT algorithm provides accurate aerosol products over dark soil, vegetation and ocean surfaces. Each algorithm has its own strengths and limitations, with DT being more limited when performing retrievals over brighter surfaces.

Response: The discussion of the DT and DB algorithms has been removed from the results section of the article, so this paragraph has been deleted

32. Figure 6: Author need to clarify collocation criteria for the satellite aerosol products and ground-based PM10 observation.

Response: The collocation criteria are discussed in the last paragraph of Section 2.3.

33. Line 494 "Therefore, the AAI calculation method is not a classic aerosol retrieval method.": What is the classic aerosol retrieval? The field of aerosol remote sensing employs numerous diverse approaches and algorithms for retrieving aerosol information, each with specific applications, advantages, and limitations.

Response: Yes, I have deleted this inaccurate expression.

34. Lines 519-520 "However, due to the limitation of the satellite observation range, Himawari-8 cannot effectively monitor dust activities in central and western Xinjiang, China.": It would like to recommend revising it to "Due to the limited field of view from geostationary orbit, Himawari-8 has reduced observational capability in central and western Xinjiang, China."

Response: Thank you for your suggestion. We have revised this sentence in Lines 536-538.

35. Lines 562-563 "However, the performance of the Himawari-8 AOD product was worse than that of several other products.": It is hard to tell from the results shown here.

Response: In the latest version of the manuscript, this conclusion is no longer drawn.

36. Reference: Suggest reviewing the manuscript to ensure that the references are appropriately used.

Response: Regarding the references, we have made careful modifications and verifications

---

## Editor Decision (ED1)

Additional Editor Comments:

Line 18. This statement implies that the Absorbing Aerosol Index (AAI) detects aerosols beneath clouds. That is incorrect. The AAI can detect UV-absorbing aerosols mixed with clouds but not beneath clouds. The AAI also detects UV-absorbing aerosols above bright backgrounds such as snow/ ice covered surfaces, or above cloud decks. This is amply documented in the literature [Herman et al., 1997; Torres et al., 1998].

Line 62. Provide references for the UV-absorption technique that has been applied to multiple sensors: TOMS (on Nimbus-7 and Earth Probe platforms, Torres et al 2002), and currently applied to OMI on Aura platform (Torres et al., 2007), EPIC on DSCOVR satellite (Ahn et al, 21), and TROPOMI on Sentinel5-P (Torres et al., 2020). There are many instances or unresolved acronyms such as TOMS (Total Ozone Mapping Spectrometer) and others.

Line 68. Add reference for the NASA TROPOMI aerosol product (Torres et al., 2020).

Line 76. The earliest analysis of this type was carried out by Prospero et al (2002) based on TOMS data.

Line 196. Be consistent in the naming of the sub-sections 2.2.1 through 2.2.4. Currently 2.2.1 refers to satellite, 2.2.2 to the sensor, while2.2.3 and 2.2.4 refer again to the satellite. I suggest using hyphenated sensor-satellites such as MODIS-TERRA/AQUA, TROPOMI/S5P, etc.

Line 216. Clarify that the UVAI method that detects aerosol mixed with clouds and above cloud-layers is only a qualitative measurement. In the last twenty years, however, accurate approaches to derive quantitative information such as aerosol optical depth and single scattering albedo of smoke and dust aerosols for cloud-free conditions (Torres et al, 2002, 2007; Ahn et al, 2021) and above clouds from UV (Torres et al., 2012) and visible observations (Jethva et al., 2013) have been developed.

Line 268. Revisit Table 1. I suggest including five columns: Sensor, Satellite, Product, Threshold, Reference.

Line 488. Replace cloud cover with sub-pixel cloud contamination

Cited references:

Ahn, C., Torres, O., Jethva, H., Tiruchirapalli, R., & Huang, L.-K. (2021). Evaluation of aerosol properties observed by DSCOVR/EPIC instrument from the Earth-Sun Lagrange 1 Orbit.

*Journal of Geophysical Research:Atmospheres*, *126*, e2020JD033651. https://doi.org/10.1029/2020JD033651

Herman, J.R., P.K. Bhartia, O. Torres, C.Hsu , C. Seftor, and E. Celarier, Global Distribution of UV-absorbing Aerosols From Nimbus-7/TOMS data, *J. Geophys. Res.*, 102, 16911-16922, 1997

Torres O., P.K. Bhartia, J.R. Herman and Z. Ahmad, Derivation of aerosol properties from satellite measurements of backscattered ultraviolet radiation. Theoretical Basis, *J. Geophys. Res.*, 103, 17099-17110, 1998.

Torres, O., P. K. Bhartia, J. R. Herman, A. Syniuk, P. Ginoux, and B. Holben (2002a), A long term record of aerosol optical depth from TOMS observations and comparison to AERONET measurements, J. Atmos. Sci., 59, 398– 413.

Prospero, J.M., P.Ginoux, O. Torres, and S.E. Nicholson, Environmental characterization of atmospheric soil dust derived from the Nimbus-7 TOMS absorbing aerosol product, *Reviews of Geophysics*, 10.129/2000RG000095, Sept 4, 2002

Torres, O., H. Jethva, and P. K. Bhartia (2012), Retrieval of aerosol optical depth above clouds from OMI observations: Sensitivity analysis and case studies, J. Atmos. Sci., 69, 1037–1053, doi:10.1175/JAS-D-11-0130.1.

Jethva, H., O. Torres, L. A. Remer, and P. K. Bhartia (2013), A color ratio method for simultaneous retrieval of aerosol and cloud optical thickness of above-cloud absorbing aerosols from passive sensors: Application to MODIS measurements, IEEE Trans. Geosci. Remote Sens., 51(7), 3862–3870, doi:10.1109/TGRS.2012.2230008.

---

## Author Response (AR2)

**Responses to Editor:**

1. Line 18. This statement implies that the Absorbing Aerosol Index (AAI) detects aerosols beneath clouds. That is incorrect. The AAI can detect UV-absorbing aerosols mixed with clouds but not beneath clouds. The AAI also detects UV-absorbing aerosols above bright backgrounds such as snow/ ice covered surfaces, or above cloud decks. This is amply documented in the literature [Herman et al., 1997; Torres et al., 1998].

Response: Thanks for your suggestion. We revised this sentence to: but also compensates for the shortcomings of other products that cannot detect UV-absorbing aerosols mixed with clouds.

2. Line 62. Provide references for the UV-absorption technique that has been applied to multiple sensors: TOMS (on Nimbus-7 and Earth Probe platforms, Torres et al 2002), andcurrently applied to OMI on Aura platform (Torres et al., 2007), EPIC on DSCOVR satellite (Ahn et al, 21), and TROPOMI on Sentinel5-P (Torres et al., 2020). There are many instances or unresolved acronyms such as TOMS (Total Ozone Mapping Spectrometer) and others.

Response: Thanks. In this section, we added the corresponding references.

3. Line 68. Add reference for the NASA TROPOMI aerosol product (Torres et al., 2020).

Response: Thanks. We have added the references.

4. Line 76. The earliest analysis of this type was carried out by Prospero et al (2002) based on TOMS data.

Response: Thanks. In the article, we added this reference.

5. Line 196. Be consistent in the naming of the sub-sections 2.2.1 through 2.2.4. Currently 2.2.1 refers to satellite, 2.2.2 to the sensor, while2.2.3 and 2.2.4 refer again to the satellite. Isuggest using hyphenated sensor-satellites such as MODIS-TERRA/AQUA, TROPOMI/S5P, etc.

Response: Thanks for your suggestion. We have modified the corresponding titles.

6. Line 216. Clarify that the UVAI method that detects aerosol mixed with clouds and above cloud-layers is only a qualitative measurement. In the last twenty years, however, accurate approaches to derive quantitative information such as aerosol optical depth and single scattering albedo of smoke and dust aerosols for cloud-free conditions (Torres et al, 2002, 2007; Ahn et al, 2021) and above clouds from

UV (Torres et al., 2012) and visible observations (Jethva et al., 2013) have been developed.

Response: Thanks for your suggestion. We revised this sentence to: Additionally, unlike satellite-based AOD measurements, UVAI can qualitatively characterize the presence of absorbing aerosols and their spatial distribution characteristics even under cloudy conditions, thereby achieving daily global coverage.

7. Line 268. Revisit Table 1. I suggest including five columns: Sensor, Satellite, Product, Threshold, Reference.

Response: Thanks. We have supplemented Table 1.

8. Line 488. Replace cloud cover with sub-pixel cloud contamination.

Response: Thanks. We have replaced it.